# New Cases of Hypochromic Microcytic Anemia Due to Mutations in the *SLC11A2* Gene and Functional Characterization of the G75R Mutation

**DOI:** 10.3390/ijms23084406

**Published:** 2022-04-15

**Authors:** Lídia Romero-Cortadellas, Gonzalo Hernández, Xènia Ferrer-Cortès, Laura Zalba-Jadraque, José Luis Fuster, Mar Bermúdez-Cortés, Ana María Galera-Miñarro, Santiago Pérez-Montero, Cristian Tornador, Mayka Sánchez

**Affiliations:** 1Iron Metabolism: Regulation and Diseases, Department of Basic Sciences, Universitat Internacional de Catalunya (UIC), 08195 Sant Cugat del Vallès, Spain; lromerocor@uic.es (L.R.-C.); ghernandezv@uic.es (G.H.); xferrerc@uic.es (X.F.-C.); 2BloodGenetics S.L. Diagnostics in Inherited Blood Diseases, 08950 Esplugues de Llobregat, Spain; laurazalbaj@gmail.com (L.Z.-J.); sperez@bloodgenetics.com (S.P.-M.); ctornador@bloodgenetics.com (C.T.); 3Pediatric OncoHematology Service, Clinic University Hospital Virgen de la Arrixaca, Instituto Murciano de Investigación Biosanitaria (IMIB), 30120 Murcia, Spain; josel.fuster@carm.es (J.L.F.); mariam.bermudez2@carm.es (M.B.-C.); anam.galera@carm.es (A.M.G.-M.)

**Keywords:** DMT1, microcytic anemia, iron overload, *SLC11A2*, mutation, EPO

## Abstract

Divalent metal-iron transporter 1 (DMT1) is a mammalian iron transporter encoded by the *SLC11A2* gene. DMT1 has a vital role in iron homeostasis by mediating iron uptake in the intestine and kidneys and by recovering iron from recycling endosomes after transferrin endocytosis. Mutations in *SLC11A2* cause an ultra-rare hypochromic microcytic anemia with iron overload (AHMIO1), which has been described in eight patients so far. Here, we report two novel cases of this disease. The first proband is homozygous for a new *SLC11A2* splicing variant (c.762 + 35A > G), becoming the first ever patient reported with a *SLC11A2* splicing mutation in homozygosity. Splicing studies performed in this work confirm its pathogenicity. The second proband harbors the previously reported DMT1 G75R mutation in homozygosis. Functional studies with the G75R mutation in HuTu 80 cells demonstrate that this mutation results in improper DMT1 accumulation in lysosomes, which correlates with a significant decrease in DMT1 levels in patient-derived lymphoblast cell lines (LCLs). We also suggest that recombinant erythropoietin would be an adequate therapeutic approach for AHMIO1 patients as it improves their anemic state and may possibly contribute to mobilizing excessive hepatic iron.

## 1. Introduction

Divalent metal transporter 1 (DMT1) is a proton-coupled, transmembrane protein encoded by the *SLC11A2* gene and located in the brush border of duodenal enterocytes, where it ensures dietary non-heme iron absorption from the apical membrane to the cytosol. In other cells, especially in erythroid precursors, DMT1 mediates iron transport from acidified endosomes to the cytosol, coordinately with transferrin receptor-dependent iron uptake [1].

Four distinct DMT1 isoforms exist due to alternative transcription initiation sites and alternative RNA splicing at the 3′ end. The 1A and 1B isoforms differ in the N-terminal region, and the alternative splicing in the 3′ end is responsible for two additional isoforms with different C-terminal and 3′UTR regions, one of them including an iron responsive element (IRE) [2]. These DMT1 isoforms are tissue-specific and show functional differences in protein trafficking, accounting for their different roles in iron homeostasis. While DMT1-IRE is mostly expressed in the duodenum, DMT1-nonIRE has a broader tropism although it is especially important in erythroid precursors. In the cell, both isoforms are initially targeted to the plasma membrane and internalized to early endosomes through chlatrin-dependent (DMT1-nonIRE) or chlatrin-independent (DMT1-IRE) mechanisms. Subsequently, DMT1-nonIRE returns to the membrane together with the transferrin receptor via recycling endosomes. Instead, DMT1-IRE is mainly sorted to late endosomes and lysosomes [3].

Eight patients suffering from the rare disease hypochromic microcytic anemia have been described so far to be harboring mutations in the *SLC11A2* gene (OMIM #206100; Anemia, hypochromic microcytic, with iron overload 1, AHMIO1). A summary of all published cases is described in Table 1. Notably, all individuals but two presented with liver iron overload, as assessed by magnetic resonance imaging (MRI) and/or liver biopsy.

Here, we report two new patients with hypochromic microcytic anemia due to mutations in the *SLC11A2* gene, being the ninth and tenth world-wide identified cases, and we study the pathophysiologic consequences of the DMT1 G75R mutation.

## 2. Results

### 2.1. Family 1

#### 2.1.1. Clinical Data 

In family 1, proband II.2 is a 1-year-old girl born from consanguineous parents of Moroccan origin (Figure 1A). At birth, she presented with severe microcytic anemia and intense erythroblastemia, together with high serum iron levels, elevated transferrin saturation and normal serum ferritin (Table 2). Anemia was transfusion-dependent until she was 6 months old; then, a significant improvement in hemoglobin levels was detected in subsequent tests. Oral iron supplements were administered until eight months of age together with vitamin D3. Bone marrow aspirate showed normal cytology except for megakaryocyte and erythroid hyperplasia without dysplasia. She also presented non-obstructive hypertrophic myocardiopathy with a tendency to resolve after the correction of anemia. Liver iron levels could not be assessed as the patient is too young. 

#### 2.1.2. *SLC11A2* c.[762 + 35A > G] Variant Causes Aberrant Splicing

A targeted next generation sequencing (NGS) panel performed in patient 1-II.2 identified an homozygous A > G substitution at position +35 bp within intron 8 in the *SLC11A2* gene (NM_000617.3:c.[762 + 35A > G];[762 + 35A > G]), which was confirmed by conventional Sanger sequencing (Figure 1C). The mother and brother were healthy carriers, while the father was not available for analysis. This variant is absent from public databases. 

The effect of the c.[762 + 35A > G] mutation was first predicted with Mutation Taster software, which showed the creation of a potential new donor splice site 35 bp after the original splice donor site. This aberrantly spliced mRNA would result in the inclusion of an intronic region in the predicted *SLC11A2* CDS, which would ultimately result in a premature stop codon and, consequently, a truncated protein (Figure 2A, top). 

Splicing functional studies were performed with the patient’s peripheral blood mononuclear cells (PBMCs) treated with or without puromycin, a translation inhibitor that suppresses the nonsense-mediated decay machinery (NMD). Results revealed an extra band of 287 bp present in the patient and in the carrier relatives’ samples, but it was absent in the control sample (Figure 2A, bottom). The *SLC11A2* pathogenic allele could still be detected in patient and relatives’ untreated cells, suggesting that the aberrant mRNA is not completely degraded by NMD machinery. Sequencing the mutated band confirmed the introduction of 34 nucleotides from intron 8 of the *SLC11A2* gene (Figure 2B). 

*SLC11A2* mRNA levels in patient PBMCs were half the ones observed in healthy controls, which were further restored by the addition of puromycin (Figure 2C). 

### 2.2. Family 2

#### 2.2.1. Clinical Data

In family 2, proband II.4 is a 3-year-old boy born from non-consanguineous parents of Ecuadorian origin. This family had a previous child that died intrauterine at week 39 presenting cardiomegaly and fetal ascites. Proband 2-II.4 suffered from fetal cardiomegaly and anemia at the neonatal period, requiring four intrauterine transfusions. Bone marrow morphologic examination showed dyserythropoiesis traits and no evidence of pathological sideroblasts on Perls Prussian blue staining. Hemoglobin levels were normal at birth, probably as a result of intrauterine transfusions. Hypochromic microcytic anemia was present at one month of age, with a concomitant increase in serum iron levels and transferrin saturation (Table 2). He presented sustained high serum iron levels as well as transferrin saturation, with normal-to-low ferritin levels. 

The patient received periodic transfusions every three to six months until he was two and a half years old. At that point, recombinant erythropoietin treatment (rEPO, administered subcutaneously) was started at 0.64 µg/kg every two weeks. The erythropoietin dosage was progressively increased over time, and it is currently established at 3.28 µg/kg every two weeks. Since the treatment was established, improved but sub-normal hemoglobin levels have been attained (~10 g/dL). Liver iron overload was assessed by MRI at the age of two years, but the results were inconclusive due to artifacts associated with sedation. 

#### 2.2.2. DMT1 Is Destabilized by the G75R Mutation

The homozygous *SLC11A2* pathogenic mutation NM_000617.3:c.[223G > A];[223G > A] in exon 4 was identified in patient 2-II.4 by using a targeted NGS panel. The parents and brother were healthy carriers of the variant, while the patient’s sister was a non-carrier (Figure 1B,D). This change results in the protein substitution NP_000608.1:p.[G75R];[G75R], located in the first transmembrane domain. G75R mutation has been previously reported as a pathogenic variant in two unrelated patients [8,11]. 

The predicted structure of human DMT1 was obtained from the AlphaFold database. DMT1 Glycine75 is a highly conserved amino acid among multiple species (Figure 3A) and is located in transmembrane domain 1 according to TMHMM. The structural computational model shows that the G75R mutation leads to the steric collision of Arginine75 with Glutamic acid208 from transmembrane domain 5 (Figure 3B). 

#### 2.2.3. G75R Mutation Causes DMT1 Accumulation in the Lysosomes

Duodenal HuTu 80 cells were transiently transfected with *SLC11A2* constructs bearing the DMT1-IRE and -nonIRE isoforms of wild-type (WT), G75R, G212V and N491S DMT1 variants in (Appendix A). The subcellular localization of DMT1 was analyzed by immunofluorescence-based co-localization using antibodies against calnexin (to label endoplasmic reticulum), EEA1 (to label early endosomes) and LAMP1 (to label late endosomes/lysosomes). The results showed that, in agreement with previously reported data [9], the N491S mutation causes protein retention in the ER, although only for the IRE form. As expected, G212V substitution showed a similar subcellular localization relative to the WT form (Figure 4A). In addition, all nonIRE isoforms significantly co-localized with early endosomes compared to their IRE form counterparts (Appendix A), while the IRE forms were mostly located in late endosomes/lysosomes (Appendix A). However, G75R mutated isoforms do not co-localize with the ER (Figure 4A, Appendix A). Instead, a significant increase in both DMT1-IRE and -nonIRE isoforms of the G75R mutant version is detected in late endosomes/lysosomes compared to the WT forms (Figure 4B). 

In line with these results, DMT1 protein levels were significantly decreased (~55%) in proband 2-II.4 lymphoblastoid cell lines (LCLs) compared to controls (Figure 1D). This decrease in DMT1 levels is not caused by mRNA degradation as the *SLC11A2* mRNA expression levels in patient LCLs did not differ from those observed in control LCLs (Figure 1C). Overall, the data suggest that the G75R mutation results in protein degradation due to increased lysosomal targeting. 

Mutations in the *SLC11A2* gene have been described in eight patients since the first reported case in 2005 [4,5]. Although, initially, the disease was defined as hypochromic microcytic anemia with liver iron overload (AHMIO1), two out of the eight reported cases did not present with the iron-loaded hepatic phenotype. In this study, we have presented two new AHMIO1 patients, one of them harboring the first homozygous splicing mutation described in *SLC11A2*. This variant creates a new splicing donor and, in turn, an aberrant mRNA, which undergoes partial degradation by NMD as it is still detectable in patient-derived PBMCs [14]. In fact, *SLC11A2* mRNA can be detected in patient 1-II.2 PBMCs although its levels are significantly decreased compared to healthy donors. As a consequence, a truncated DMT1 protein is expected to be synthesized and possibly degraded; unfortunately, due to limited sample availability, DMT1 levels could not be verified in this patient. The finding of this mutation might be important for molecular diagnostic approaches in patients with hypochromic microcytic anemia since deep intronic regions are not commonly analyzed. 

The second patient is a homozygote for the DMT1 G75R mutation, previously described in two other independent Ecuadorian cases. As our case is also of Ecuadorian origin, we hypothesize a possible founder effect for this mutation. The results presented in this study indicate that this mutation would affect DMT1 stability and, probably, its insertion in the plasma membrane. In fact, mutations involving polar, ionizable residues (notably arginine) are more often associated with protein malfunction and disease when located in transmembrane domains rather than in cytosolic domains [15]. 

Taking into account the modeling results, we hypothesized that G75R mutation may alter DMT1 trafficking and/or subcellular localization. The hereby presented co-localization experiments of several DMT1 mutants with various organelles showed an enrichment of nonIRE forms in early endosomes, while all IRE isoforms co-localized more with late endosomes/lysosomes, in line with what has been previously reported [2,3]. As previously reported by Bardou-Jacquet and collaborators, we observed an increased co-localization of N491S with ER, while the G212V mutant showed a similar subcellular distribution to the WT counterpart [9]. Notably, our results also showed a significant accumulation of G75R mutants in late endosomes/lysosomes. Furthermore, DMT1 levels in patient 2.II.4-derived LCLs were reduced, while *SLC11A2* mRNA levels remain unaffected in these cells. These results overall suggest that the G75R variant enhances DMT1 lysosomal degradation. 

Formally, we cannot exclude the possibility that the G75R mutation may also affect the transport of iron through DMT1. The Belgrade anemic rat bearing the analogous G185R mutation to the human G75R mutation presented with lower duodenal DMT1 protein levels, and that was attributed to accelerated protein degradation or impaired release of the mutant protein from its site of synthesis [16]. The latter is not the case in the human G75R mutation as we did not observe retention in the ER. 

The injection of iron dextran in the Belgrade rat (DMT1 G185R) improved its anemic state but led to hepatic iron deposition [17,18]. This DMT1-independent iron acquisition, in a high serum iron level condition, is performed probably through the ZIP14 transporter [19]. So far, all published cases but two with mutations in the *SLC11A2* gene presented with liver iron overload. Therefore, we suggest that liver iron accumulation observed in some AHMIO1 patients might be exacerbated by transfusions and/or oral iron supplements, the gold standard treatments for general anemic patients. This highlights the importance of a proper and early diagnosis for AHMIO1 patients to avoid counterproductive treatments. As seen with our proband 2-II.4 and in two additional cases from the literature (Table 1), we foresee that treatment with recombinant EPO is an appropriate therapeutic approach for AHMIO1 patients as it helps raise hemoglobin levels by possibly mobilizing excessive hepatic iron.

## 3. Materials and Methods

### 3.1. DNA Extraction, PCR Amplification and DNA Sequencing

Patient DNA was extracted from peripheral blood using the QIAamp DNA Blood Mini Kit (Qiagen, Redwood City, CA, USA) and analyzed using a targeted NGS gene panel (v16 #10030 for proband 1-II.2 and v15 for proband 2-II.4) in BloodGenetics S.L. (Esplugues de Llobregat, Spain), which included the following genes: *ACVR1*, *ATP4A*, *ATP7B*, *TF*, *CP*, *TMPRSS6*, *SLC11A2* and *STEAP3*. The library was constructed using the Custom HaloPlex^TM^ HS Target Enrichment System (Agilent Technologies, Santa Clara, CA, USA) and sequenced on a MiniSeq platform (Illumina, San Diego, CA, USA) [20]. Data were analyzed with SureCall software (Agilent Technologies, Santa Clara, CA, USA) and Varsome Clinical software (Saphetor SA, Lausanne, Switzerland) [21]. PCR validation of the specific mutations was performed with 50 ng of genomic DNA. Primer sequences and PCR conditions are available upon request. The resulting amplification products were verified on a 2% ethidium bromide agarose gel. PCR products were purified with NucleoSpin^®^ Gel and PCR Clean-up Kit (Macherey-Nagel, GmbH & co KG, Düren, Germany) and then sequenced using the conventional Sanger method. Sequencing results were analyzed using Chromas 2.6.6 (Technelysium Pty Ltd., South Brisbane, Australia) software.

Genetic variants refer to NM_000617.3 for the *Homo sapiens SLC11A2* transcript variant, and NP_000608.1 for *Homo sapiens* DMT1 protein.

### 3.2. Bioinformatics and Computational Analysis

Mutation Taster was used to predict the putative splicing effects of the *SLC11A2* c.762 + 35A > G (NM_001174125.2) variant [22]. DMT1 sequences from different species were retrieved from NCBI or UNIPROT and aligned with Clustal Omega for multiple alignments. DMT1-predicted structure was obtained from AlphaFold, as no crystallographic structure was available [23,24]. Transmembrane domain topology prediction was performed with TMHMM Server (RRID_SCR_014935) [25]. Mutagenesis and analysis were performed with the PyMOL Molecular Graphics System software, version 4.6 (Schrödinger, LLC, New York, NY, USA). The rotamer with the highest score was selected. The prediction of the putative effects of the variant in the structure/function of the protein was performed by visual inspection. 

### 3.3. Plasmids and Mutagenesis

The plasmid constructs pdsRed2-C1 DMT1 WT-IRE, WT-nonIRE, G212V-IRE, G212V-nonIRE, N491S-IRE and N491S-nonIRE, which included the sequences into the Xho1–Hind3 site of the pdsRed2-C1 expression vector (Clontech Laboratories Inc., Takara Bio, Kusatu, Shiga Prefecture, Japan), were generous gifts from Dr. Edouard Bardou-Jacquet and Dr. Olivier Loréal. All of them included the 1B isoform of DMT1. Site-directed mutagenesis was performed on pdsRed2-C1 DMT1 WT-IRE and WT-nonIRE to obtain p.G75R by a standard protocol with the following specific primers: Fw 5′ ACTCTGGGCTTTCACCAGACCAGGTTTTCTATG 3′ and Rv 5′ CATAGAAAACCTGGTCTGGTGAAACCCGA 3′.

### 3.4. Cells and Cell Lines

HuTu 80 (ATCC^®^ HTB-40TM) cells were grown in Eagle’s Minimum Essential medium (ATCC^®^ 30-2003TM) supplemented with 10% fetal bovine serum (FBS), 1% L-glutamine, 1% antibiotics (penicillin/streptomycin).

Peripheral blood mononuclear cells (PBMCs) were isolated from the patients, family members and healthy unrelated controls from EDTA-treated blood by Lymphoprep^TM^ (StemCell Technologies, Vancoucer, BC, Canada) density gradient, following the manufacturer’s instructions. PBMCs were grown in RPMI-1640 with 15% heat inactivated FBS, 2 mM L-glutamine, 50 µg/mL streptomycin, 100 U/mL penicillin and 2.5 µg/mL phytohemagglutinin-L (ThermoFisher Scientific, Waltham, MA, USA) for 7 days. EBV-transformed and immortalized Lymphoblastoid Cell Lines (LCLs) from proband 2-II.4 and healthy unrelated controls were established as described in [26]. Puromycin (75 µL/mL) (Sigma-Aldrich, St. Louis, MO, USA) was added 5 h before RNA extraction.

### 3.5. RNA Extraction, Reverse Transcription and qPCR

Total RNA was extracted from PBMCs (patient 1-II.2) and LCLs (patient 2-II.4) with TRIzol^TM^ Reagent (ThermoFisher Scientific, MA, USA) following the manufacturer’s instructions. Five hundred ng of total RNA was reverse transcribed in a 20 µL reaction using GoScript^TM^ Reverse Transcriptase (Promega, Madison, WI, USA) with random primers, following the manufacturer’s instructions.

Twenty ng of cDNA was subjected to real time quantitative PCR in the CFX96 Real -Time System C1000 Touch Thermal Cycles detection system using the SYBR(R) Green Master mix (both from Bio-Rad, Hercules, CA, USA). Quantitative RT-PCR for *SLC11A2* mRNA expression was performed using the following primer sets: *SLC11A2* Fw TCCATTCCTGAGGAGGAGTA and Rv CAGACTGCAAATCGGATTCA; *GAPDH* Fw ATGGGGAAGGTGAAGGTCG and Rv GGGGTCATTGATGGCAACAATA. For each gene, patient and control (n = 3) samples were analyzed in the same RT-PCR and run in triplicate in three independent experiments. Relative quantification (RQ) was calculated with the 2-∆∆Ct method using *GAPDH* as a housekeeping gene. *SLC11A2* mRNA expression was compared to that of the pool of three controls in each independent experiment. Control PBMCs or LCLs were used according to the available patient sample. 

### 3.6. Splicing Studies

For family 1, the amplification of *SLC11A2* from PBMCs’ cDNA was performed using the following pair of primers: Fw 5′ GACAAATATGGCTTGCGGAAG 3′ and Rv 5′ TCCGGTTTACCTGTCTAGACTT 3′. About 40 ng cDNA (1.5 microliters), 1× PFU buffer, 0.3 mM dNTPs, 2 mM MgCl2 and 0.6 µL Pfu were used in each PCR reaction in a 50-µL reaction mixture. After initial denaturation at 95 °C for 2 min, amplification was performed for 30 cycles at 95 °C for 30 s, 60 °C for 30 s and 72 °C for 30 s, then at 72 °C for 5 min. PCR products were analyzed on 3% agarose gels and then purified with NucleoSpin^®^ Gel and PCR Clean-up Kit (Macherey-Nagel, GmbH & co KG, Düren, Germany). Sanger sequencing was performed to confirm that the variants and peaks were separated by CRISP-ID (v1.1).

### 3.7. Immunofluorescence

HuTu 80 cells were seeded in 24-well plates and transiently transfected with 500 ng DNA using Lipofectamine^TM^ 3000 (ThermoFisher Scientific, Waltham, MA, USA) following the manufacturer’s instructions. At 48 h posttransfection, cells were fixed in 4% paraformaldehyde, permeabilized with 0.1% Triton X-100 (calnexin and EEA1) or 0.05% saponin (LAMP1), blocked with PBS + 2% BSA (only calnexin and EEA1) and incubated 1 h at 37 °C or room temperature (LAMP1) with the appropriate antibodies: rabbit anti-calnexin (1:400) (ab22595, Abcam, Cambridge UK) to label endoplasmic reticulum; rabbit anti-EEA1 (1:1000) (ab2900, Abcam, Cambridge, UK) to label early endosomes; rabbit anti-LAMP1 to label late endosomes/lysosomes (1:400) (L1418, Sigma-Aldrich, St. Louis, MO, USA). Cells were then washed and incubated 1 h at room temperature with an Alexa Fluor^®^ 488-tagged secondary antibody (goat anti-rabbit, 1:300, ThermoFisher Scientific, MA, USA). Coverslips were washed with 1× PBS and mounted on glass slides with Fluoromount-G(R) mounting media (Southern Biotech, Birmingham, AB, USA). 

### 3.8. Confocal Microscopy

Confocal imaging and quantification were performed as described in [27]. Approximately 20 stacks were acquired per each imaged cell with a Leica TCS SP8 confocal microscope using the 63×/1.40 NA oil high resolution objective. A set of two different experiments was performed, and 25 single cells from each condition were analyzed for each round (n = 50). All quantifications were performed on unmodified reconstructed 3D images with Imaris 9.2 Software (Oxford instruments, Abingdon, UK). As it is recommended for surface organelles such as ER or Golgi, for the measurement of DMT1 intensity in the ER, the mean intensity of DMT1 in ER-labelled cells was quantified using the Surface tool of the Imaris software. Instead, the Spot tool was used for the co-localization analysis of DMT1 in endosomes and late endosomes/lysosomes. The same settings for equal conditions and throughout experiments were used.

### 3.9. Immunoblotting

LCLs were washed once with PBS and then resuspended in a lysis buffer containing 20 mM Tris-HCl (pH 7.4), 5 mM EDTA (pH 8), 1% NP40, 150 mM NaCl and protease inhibitors (A32955, ThermoFisher Scientific, Waltham, MA, USA). Then, samples were sonicated and after a 5 min centrifugation at 4 °C protein concentration in the supernatant was determined by Bradford assay (Bio-Rad, Hercules, CA, USA). Forty µg total protein lysate was separated by sodium dodecyl sulfate–polyacrylamide gel electrophoresis (SDS-PAGE) (10%) and transferred to polyvinylidene fluoride (PVDF) membranes using the iBlot^TM^ 2 Transfer Device (ThermoFisher Scientific, Waltham, MA, USA). Samples were incubated for 5 min at room temperature in Laemmli buffer (containing 125 mM Tris-HCl pH 6.8, 10% SDS, 25% glycerol, 1% beta-mercaptoethanol and bromophenol blue) prior to SDS-PAGE as the heat treatment has been described to cause DMT1 protein aggregation. As a consequence, a DMT1 glycosylation pattern was conserved and, therefore, visible in the blots. 

The transfer of proteins was verified by staining the blots with Ponceau S (Sigma-Aldrich, St. Louis, MO, USA). Immunoblots were incubated with blocking solution (5% skim milk, 0.1% Tween 20, in TBS -TBST-) for one hour at room temperature before incubation with primary antibodies for 18 h at 4 °C in blocking solution. Primary antibodies included rabbit anti-DMT1 (1:1000, #15083 Cell Signalling Technology, Danvers, MA, USA) and mouse anti-GAPDH (1:30,000, 60004-1-Ig Proteintech, Manchester, UK), the latter used as a loading control. Then, the blots were washed with TBST and incubated during one hour at room temperature with the corresponding horseradish peroxidase (HRP)–labeled secondary antibody in blocking solution: 1:5000, peroxidase conjugated anti-rabbit IgG [H + L] and 1:30,000 peroxidase conjugated anti-mouse IgG [H + L]; both from Jackson ImmunoResearch. Immunodetection was obtained using Immobilon(R) Forte Western HRP Substrate (Merck Millipore, Burlington, MA, USA), and images were taken with ChemiDoc^TM^ and quantified using the software Image Lab (both from Bio-Rad, Hercules, CA, USA). Six independent experiments were carried out, in which patient 2-II.4 and six control samples were included. The same controls were used across the experiments. DMT1 expression in the patient was expressed relative to the mean of the six controls for each experiment. 

### 3.10. Statistical Analysis

Data are represented in the figures as bar plots of the group mean, with error bars denoting the SD. Normality was assessed by the Kolmogorov–Smirnov test. Comparisons between two groups were performed using Student’s *t*-test, while the nonparametric Mann–Whitney U test was applied for those pairs in which data were not normally distributed. Multiple -IRE and -nonIRE comparisons were performed using one-way ANOVA or the non-parametric Kruskal–Wallis test. Tukey’s and Dunn’s multiple comparison tests were applied as post hoc after ANOVA and Kruskal–Wallis tests, respectively. Differences were considered significant at *p* < 0.05. Statistical analysis was performed using GraphPad Prism version 9.0 for Windows (GraphPad Software Inc., San Diego, CA, USA).

## Figures and Tables

**Figure 1 ijms-23-04406-f001:**
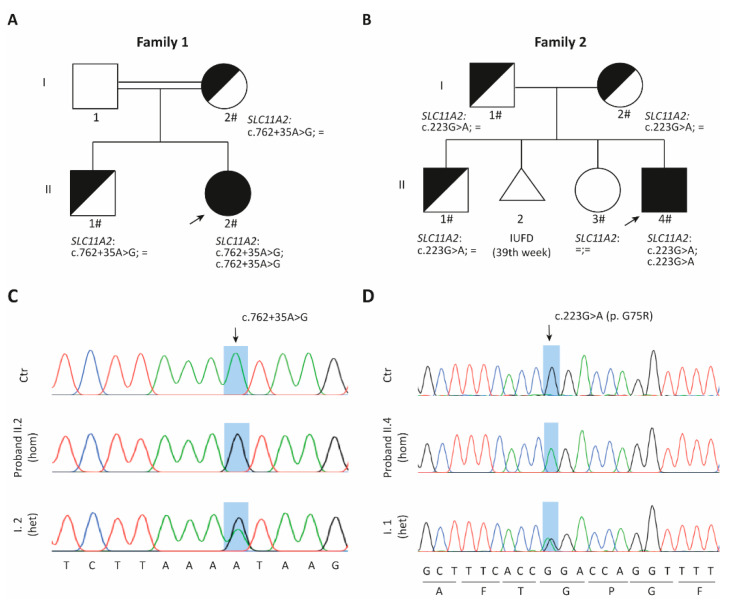
Identification of two new AHMIO1 patients. (**A**,**B**) Pedigrees of the two reported families. The probands are indicated with an arrow. Black symbols denote affected individuals, half-filled black symbols denote unaffected carriers. Individuals studied at the molecular level are indicated with the symbol #. IUFD: intrauterine fetal death. (**C**,**D**) Sanger sequencing validating the 762 + 35A > G (**C**) and 223G > A (**D**) mutations. The sequence of the respective probands is displayed, as well as that of a control (Ctr) and a relative (1-I.2 and 2-I.1, respectively). The reference sequences used are based on NM_000617.3 for *SLC11A2* and on NP_000608.1 for DMT1. Hom: homozygous. Het: heterozygous.

**Figure 2 ijms-23-04406-f002:**
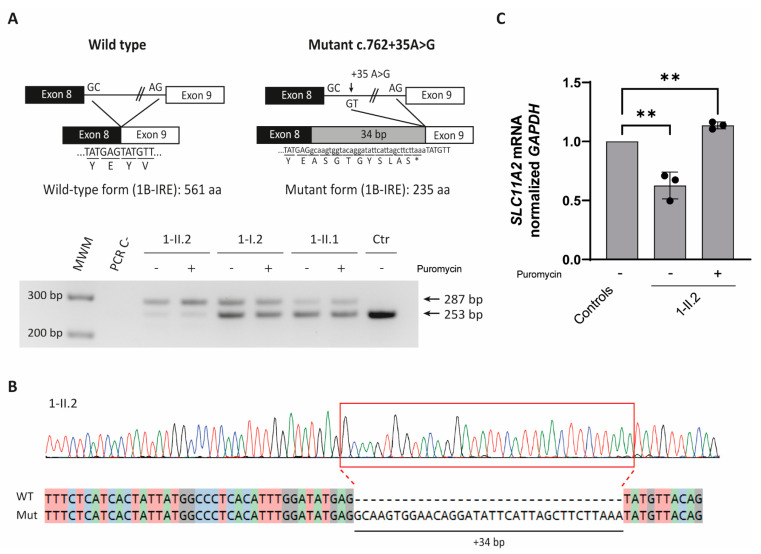
Splicing studies in proband 1-II.2. (**A**) (Top) Schematic representation of the effect of the c.762 + 35A > G mutation on *SLC11A2* mRNA and DMT1 protein. Capital letters denote coding nucleotides and lowercase letters denote intronic sequences. Codons are underlined, and the encoded amino acid is shown below. The c.762 + 35A > G splicing mutation leads to the inclusion of 34 intronic nucleotides. The introduced stop codon is indicated with an asterisk (*). (Bottom) RT-PCR analysis in RNA from PBMCs (homozygous for the c.762 + 35A > G mutation), relatives 1-I.2 and 1-II.2 (heterozygous for the c.762 + 35A > G mutation) and a control (Ctr) in the absence or presence of puromycin (P). (**B**) Sanger sequencing analysis of amplified cDNA belonging to patient 1-II.2 from the mutation region shows the insertion of 34 extra bp. (**C**) PBMCs’ *SLC11A2* mRNA quantification is expressed relative to the expression of three healthy, unrelated donors in three independent experiments. *SLC11A2* expression levels were normalized to the control gene GAPDH. Ref: reference sequence. WT: wild type. Mut: mutant. *M*_W_M: molecular weight marker. ** *p* < 0.01.

**Figure 3 ijms-23-04406-f003:**
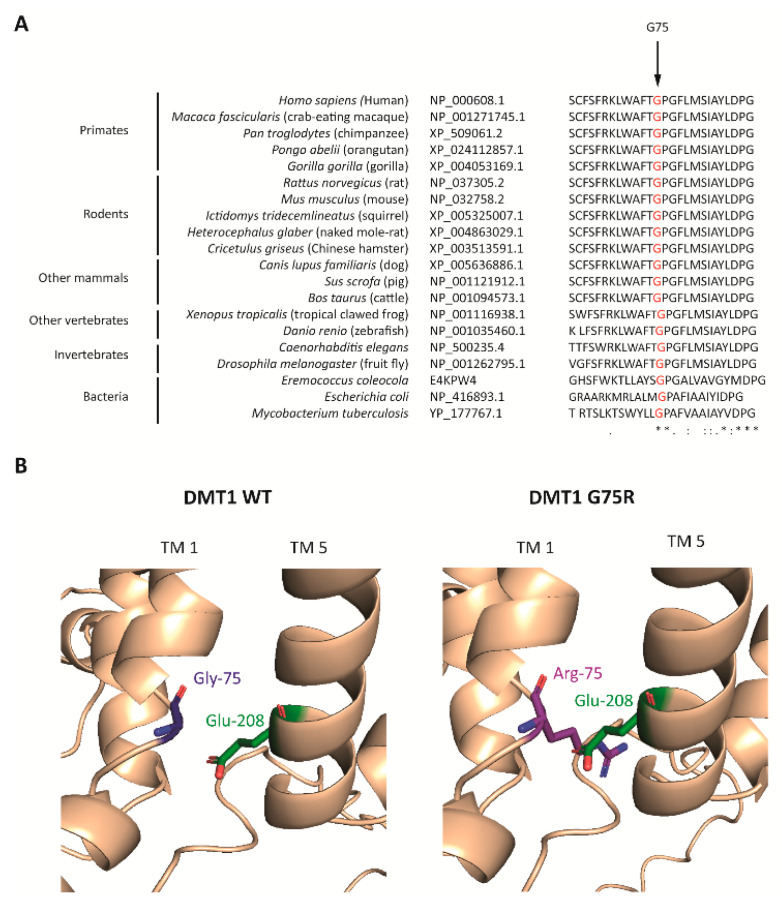
DMT1 G75 residue analysis. (**A**) Partial amino acid sequence alignment of DMT1 protein in 20 species in the vicinity of DMT1 Glycine75 (G75), indicated with an arrow. RefSeq accession numbers are reported for each sequence except for *E. coleocola*, for which a Uniprot reference is given. Below, the alignment a star (*) indicates 100% conservation of the amino acid, while colons and dots indicate amino acids with strong and weak similarity, respectively. (**B**) Structural model of the DMT1 G75R mutation. The reference (**left**) and the mutated (**right**) residues are indicated as Gly-75 and Arg-75, respectively. TM: transmembrane domain.

**Figure 4 ijms-23-04406-f004:**
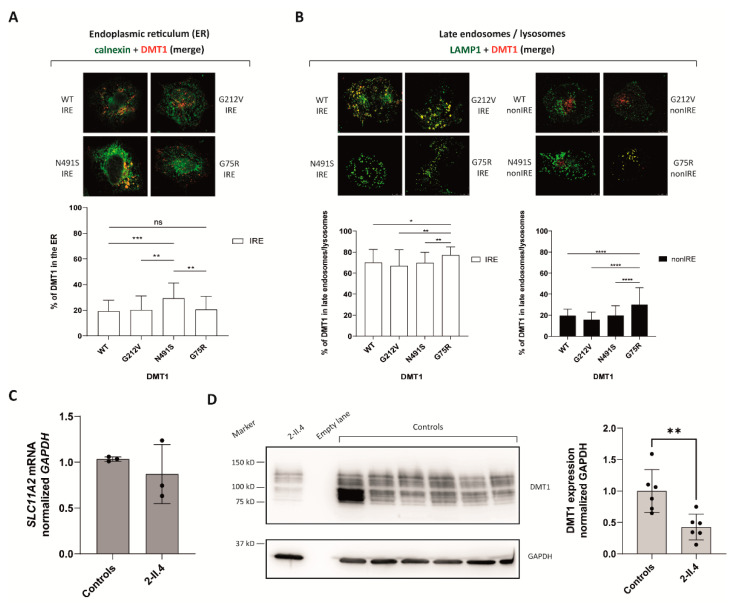
Functional characterization of G75R mutation. (**A**,**B**) Immunofluorescence studies of several DMT1 mutants in the HuTu 80 cell line. The figure shows a merged image representative of each condition, and split images are available in Appendix A. A total of 50 cells were analyzed in two independent experiments. Co-localization is expressed as the percentage of DMT1 surface in the endoplasmic reticulum (ER) (**A**) or late endosomes/lysosomes (**B**). (**C**) *SLC11A2* mRNA quantification in LCLs from proband 2-II.4 is expressed relative to the expression of three healthy, unrelated donors in three independent experiments. *SLC11A2* expression levels were normalized to the housekeeping gene GAPDH. (**D**) Representative Western blot analysis of LCLs’ lysates (40 µg) from proband 2-II.4 and six healthy, unrelated controls in six independent experiments. GAPDH was used as the loading control. The different bands reflect core- and complex-glycosylated forms of DMT1 as previously reported in [13]. * *p* < 0.05, ** *p* < 0.01, *** *p* < 0.001, **** *p* < 0.0001.

**Table 1 ijms-23-04406-t001:** Summary of all reported cases of hypochromic microcytic anemia related to *SLC11A2* mutations.

Case	Gender	Anemia Onset	Liver Iron Overload (Age)	Treatment with rEPO	Serum Ferritin	Mutations in *SLC11A2*/DMT1	Reference
1	F	At 3 m/o	Yes (19)	Initiation age n.a.	Normal to slightly elevated	E399D + E399D; exon 12 skipping	[4,5]
2	M	At birth	Yes (5)	Initiated at 3 m/o	Elevated; normalized with rEPO treatment	c.310-3-5delCTT + R416C	[6]
3	F	At birth	Yes (6)	No	Low; normalized with iron supplements	G212V + delV114	[7]
4	M	At birth	No (7)	No	Low	G75R + G75R	[8]
			No (19)	n.a.	n.a.		Data provided by Dr. Tasso
5	F	At 13 y/o	Yes (27)	No	Elevated	G212V + N491S	[9]
6	F	Infancy	Yes (25)	No	Elevated	G212V + IVS1A + 3A > T	[10]
7	M	At birth	Yes (9)	No	Normal	G75R + G75R	[11]
8	F	At birth	No (11)	Initiated at 3 y/o	Normal	R477W + IVS4 + 1G/C	[12]
9	F	At birth	NT	No	Normal	c.762 + 35A > G + c.762 + 35A > G	This study
10	M	Fetal	ID	Initiated at 2.5 y/o	Normal	G75R + G75R	This study

rEPO: recombinant erythropoietin. m/o = months old. y/o = years old. F = female; M = male. ID = inconclusive data. NT = not tested. n.a. = not available. The age of liver iron overload determination is expressed in years.

**Table 2 ijms-23-04406-t002:** Biochemical and hematological parameters in affected subjects of the two studied families.

	at Birth			6 m/o			1 y/o			1.5 and 3 y/o		
	1-II.2	2-II.4	Ref.	1-II.2	2-II.4	Ref.	1-II.2	2-II.4	Ref.	1-II.2	2-II.4	Ref.
RBC (×10^6^/µL)	3	6.2	4.6–6	4.1	4.5	4.6–6	5.5	n.a.	4.6–6	4.6	5.6	4.6–6
Hb (g/dL)	4.4	14.3	14–16.6	7.6	6.5	9.5–13.5	9.8	n.a.	10.5–13.5	8.6	8.5	10.5–13.5
MCV (fl)	72.3	72.1	106–118	58.8	48.5	74–108	61.2	n.a.	70–86	58.6	52.5	70–86
MCH (pg)	14.3	23	34–38.4	18.4	14.5	25–35	17.8	n.a.	24–30	18.5	15.3	24–30
Reticulocytes (%)	6.99	4.3	5–25	1	0.4	5–25	2	n.a.	5–25	1.4	0.53	5–25
Serum iron (µg/dL)	213	n.a.	32.6–192.7	n.a.	196	32.6–192.7	249	185	32.6–192.7	244	206	32.6–192.7
Ferritin (ng/mL)	20	n.a.	25–200	97	52	7–140	22	36	7–140	15	81	7–140
Transferrin (mg/dL)	200	n.a.	200–360	n.a.	236	200–360	223	208	200–360	209	171	200–360
Transferrin sat (%)	74.9	n.a.	20–45	n.a.	58.2	20–45	77.7	62.3	20–45	81.7	84.2	20–45
Hepcidin (ng/mL)	n.a.	n.a.	28.5–45.7	n.a.	n.a.	28.5–45.7	2.9	18.3	28.5–45.7	n.a.	n.a.	28.5–45.7

Ref. = reference. m/o = months old. y/o = years old. RBC: red blood cell. Hb = Hemoglobin. MCV = Mean Corpuscular Volume. MHC = Mean Corpuscular Hemoglobin. n.a. = not available. Values in red denote altered parameters.

## Data Availability

The data presented in this study are available on request from the corresponding author.

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
