# Peer review of "New Cases of Hypochromic Microcytic Anemia Due to Mutations in the SLC11A2 Gene and Functional Characterization of the G75R Mutation"

_ijms, 2022, doi:10.3390/ijms23084406_

Round 1
Reviewer 1 Report
In the present manuscript Romero-Cortadellas and collaborators report two novel cases of rare hypochromic microcytic anemia with iron overload (AHMIO1) caused by mutations in the SLC11A2 gene, coding for Divalent metal transporter 1 (DMT1), a protein involved in iron absorption. They also study the pathophysiologic consequences of one of the mutations (the DMT1 G75R mutation), which was previously described in two other independent cases. The reported findings are relevant for the field, as a new mutation is described and also some clues about the pathological mechanisms of the G75R mutation are provided. However, to be accepted for publication, several points need to be clarified:
- Was PBMCs’ SLC11A2 mRNA quantification in the presence of puromycin performed in the three healthy, unrelated donors? Data shown in figure 2A suggests that the 253bp band also increases in puromycin treated samples. This would further confirm that the aberrant mRNA is not completely degraded by the NMD machinery, as the WT mRNA presented similar changes.
- Authors could also use the AlphaFold predicted structure from human DMT1 to check the position of G75 and E208.
- In the first paragraph from section “2.2.3. G75R mutation causes DMT1 accumulation in the lysosomes”, some information should be included to improve the clarity of the text for those readers not familiar with the experiments performed. For instance, authors could indicate that subcellular localization of DMT1 was analyzed by immunofluorescence using antibodies against calnexin (to label endoplasmic reticulum), EEA1 332 (to label early endosomes) and anti-LAMP1 (to label late endosomes/lysosomes).
- According to authors, relative DMT1 protein content was quantified using GAPDH as loading control. Proper quantitation by western blot requires a calibration curve to ensure that the experiment is performed under non-saturating conditions. This is particularly relevant for housekeeping proteins, which may saturate easily (due to its abundance). Indeed, evidence suggests that actin, GAPDH, and tubulin, when used as a loading control, are saturated in most experiments (Moritz CP. Tubulin or Not Tubulin: Heading Toward Total Protein Staining as Loading Control in Western Blots. Proteomics. 2017 Oct;17(20)). Therefore, authors should provide as supplemental information a calibration curve for both DMT1 and GAPDH quantitation. This can be easily performed by performing a western blot with different amounts of a control sample loaded in each lane. If either DMT1 or GAPDH are saturated under their experimental conditions, they should perform the experiment under different conditions.
Minor points:
Figure 4 legend, C and D may be partially exchanged. Please revise
It should be clearly indicated how quantification of co-localization was performed. It was done using Imaris software?
Please, in Acknowledgments provide affiliation from Dr. María Tasso
Please, provide a citation or reference for Mutation Taster software
Reviewer 2 Report
Romero-Cortadellas and co-workers identified 2 new patients affected by hypochromic microcytic anemia caused by mutations in the iron importer DMT1. The manuscript is clear and well written, and findings interesting. However, some additional experiments are required to fully support their conclusions.
MAJOR POINTS
- It is not clear why authors choose G212V and N491S mutants to be studied in parallel with G75R: please comment on the different features of the mutants and provide explanation for their experimental plan.
- Colocalization analysis is not the most appropriate approach to quantify proteins distribution among different compartments. This evaluation should be performed using more precise techniques, such as organelles isolation through fractionation and proteomic analysis or Automated quantification of subcellular protein localization (Q-SCAn).
- The findings described in the manuscript are not sufficient to prove that G75R mutant protein is degraded, as claimed by the authors. To formally prove this hypothesis, lysosomal degradation should be inhibited, and mutant proteins quantified.
- Why DMT1 is not present on the plasma membrane of cells (even the WT form)? This could be an artefact of the protocol used, which may impact also on the analysis of the cellular localization of the protein.
- The decreased amount of the G75R mutant protein is not sufficient to explain the patient’ phenotype, since haploinsufficient carriers are healthy. So, some kind of functional alterations should be envisaged. For this reason, the authors should analyze the iron-import capacity of the mutant protein.
MINOR POINTS
- RBC count is an important parameter for the proper evaluation of anemia. Please insert this parameter in Table 2.
- Legend to Figure 4: Split images are in Fig.S1 (not S3, which does not exist); moreover, C and D are inverted.
- Figure 4D: which is the band corresponding to DMT1? The expected MW is 62KDa.
- Figure 4C-D: in the graphs, the controls have no StDev and no points are indicated: do the data derive from a single sample? If yes, increase the numbers; if not, explain better. Moreover, what the different points of proband are referring to? Quantification of WB presents 6 proband points and 1 (I guess) CTRL, but the picture of the WB has the opposite ratio…please clarify.
Round 2
Reviewer 1 Report
The Authors have addressed all of my concerns with the original manuscript.
Reviewer 2 Report
The authors addressed the majority of my concerns.